# BEiT:
# BERT Pre-Training of Image Transformers

**Hangbo Bao**[†,*] **Li Dong**[‡] **, Songhao Piao**[†] **, Furu Wei**[‡]
† Harbin Institute of Technology
‡ Microsoft Research
`https://github.com/microsoft/unilm`

## Abstract

We introduce a self-supervised vision representation model **BEiT**, which stands for **B**idirectional **E**ncoder representation from **I**mage **T**ransformers. Following BERT (Devlin et al., 2019) developed in the natural language processing area, we propose a *masked image modeling* task to pretrain vision Transformers. Specifically, each image has two views in our pre-training, i.e., image patches (such as $16 \times 16$ pixels), and visual tokens (i.e., discrete tokens). We first "tokenize" the original image into visual tokens. Then we randomly mask some image patches and fed them into the backbone Transformer. The pre-training objective is to recover the original visual tokens based on the corrupted image patches. After pre-training BEiT, we directly fine-tune the model parameters on downstream tasks by appending task layers upon the pretrained encoder. Experimental results on image classification and semantic segmentation show that our model achieves competitive results with previous pre-training methods.

## 1 Introduction

Transformer (Vaswani et al., 2017) has achieved promising performance in computer vision (Dosovitskiy et al., 2020; Touvron et al., 2020). However, empirical studies show that vision Transformers require more training data than convolutional neural networks. In order to solve the data-hungry issue (Liu et al., 2021a), self-supervised pre-training is a promising solution to leverage large-scale image data. Several strands of methods have been explored for vision Transformers, such as contrastive learning (Chen et al., 2021; Xie et al., 2021), and self-distillation (Caron et al., 2021).

Concurrently, BERT (Devlin et al., 2019) has achieved great success in natural language processing. Its masked language modeling task first randomly masks some proportion of tokens within a text, and then recovers the masked tokens based on the Transformer encoding results of the corrupted text. Motivated by BERT, we turn to the denoising auto-encoding idea to pretrain vision Transformers, which has not been well studied by the vision community. It is challenging to directly apply BERT-style pre-training for image data. First of all, there is no pre-exist vocabulary for vision Transformer's input unit, i.e., image patches. So we cannot simply employ a softmax classifier to predict over all possible candidates for masked patches. In contrast, the language vocabulary, such as words and BPE (Sennrich et al., 2016), is well-defined and eases auto-encoding prediction. A straightforward alternative is regarding the task as a regression problem, which predicts the raw pixels of masked patches. However, such pixel-level recovery task tends to waste modeling capability on pre-training short-range dependencies and high-frequency details (Ramesh et al., 2021). Our goal is to overcome the above issues for pre-training of vision Transformers.

In this work, we introduce a self-supervised vision representation model **BEiT**, which stands for **B**idirectional **E**ncoder representation from **I**mage **T**ransformers. Inspired by BERT, we propose a pre-training task, namely, masked image modeling (MIM). As shown in Figure 1, MIM uses two views for each images, i.e., image patches, and visual tokens. We split the image into a grid of patches that are the input representation of backbone Transformer. Moreover, we "tokenize" the image to discrete visual tokens, which is obtained by the latent codes of discrete VAE (Ramesh et al., 2021).

---

*Contribution during internship at Microsoft.

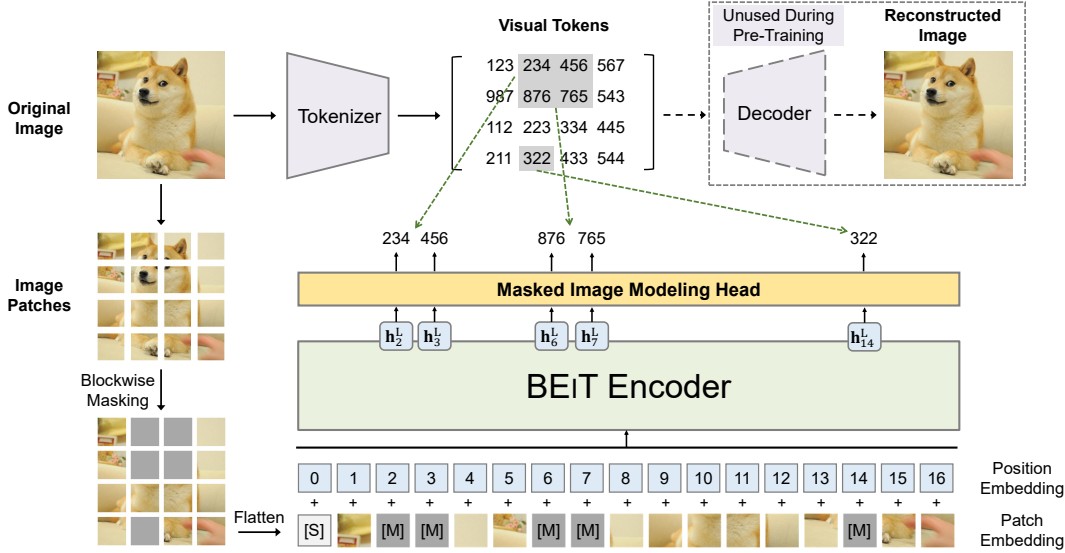

Figure 1: Overview of BEIT pre-training. Before pre-training, we learn an "image tokenizer" via autoencoding-style reconstruction, where an image is tokenized into discrete visual tokens according to the learned vocabulary. During pre-training, each image has two views, i.e., image patches, and visual tokens. We randomly mask some proportion of image patches (gray patches in the figure) and replace them with a special mask embedding [M]. Then the patches are fed to a backbone vision Transformer. The pre-training task aims at predicting the visual tokens of the *original* image based on the encoding vectors of the *corrupted* image.

During pre-training, we randomly mask some proportion of image patches, and feed the corrupted input to Transformer. The model learns to recover the visual tokens of the original image, instead of the raw pixels of masked patches.

We perform self-supervised learning and then fine-tune the pretrained BEIT on two downstream tasks, i.e., image classification, and semantic segmentation. Experimental results indicate that BEIT outperforms both from-scratch training and previous strong self-supervised models. Moreover, BEIT is complementary to supervised pre-training. Performance of BEIT can be further improved by intermediate fine-tuning with ImageNet labels. Ablation studies show that our proposed techniques are critical to the effectiveness of BERT-style pre-training for image data. Apart from performance, the improvements of convergence speed and stability of fine-tuning reduce training costs on end tasks. In addition, we demonstrate that self-supervised BEIT can learn reasonable semantic regions via pre-training, unleashing the rich supervision signals contained in images.

Our contributions are summarized as follows:

- We propose a masked image modeling task to pretrain vision Transformers in a self-supervised manner. We also provide a theoretical explanation from the perspective of variational autoencoder.

- We pretrain BEIT and conduct extensive fine-tuning experiments on downstream tasks, such as image classification, and semantic segmentation.

- We present that the self-attention mechanism of self-supervised BEIT learns to distinguish semantic regions and object boundaries, although without using any human annotation.

## 2 METHODS

Given an input image $x$, BEIT encodes it to contextualized vector representations. As shown in Figure 1, BEIT is pretrained by the masked image modeling (MIM) task in a self-supervised learning manner. MIM aims at recovering the masked image patches based on encoding vectors. For

downstream tasks (such as image classification, and semantic segmentation), we append task layers upon pretrained BEiT and fine-tune the parameters on the specific datasets.

## 2.1 IMAGE REPRESENTATIONS

The images have two views of representations in our method, namely, *image patch*, and *visual tokens*. The two types serve as input and output representations during pre-training, respectively.

### 2.1.1 IMAGE PATCH

The 2D image is split into a sequence of patches (Dosovitskiy et al., 2020), so that a standard Transformer can directly accept image data. Formally, we reshape the image $x \in \mathbb{R}^{H \times W \times C}$ into $N = HW/P^2$ patches $x^p \in \mathbb{R}^{N \times (P^2 C)}$, where $C$ is the number of channels, $(H, W)$ is the input image resolution, and $(P, P)$ is the resolution of each patch. The image patches $\{x_i^p\}_{i=1}^N$ are flattened into vectors and are linearly projected, which is similar to word embeddings in BERT (Devlin et al., 2019). Image patches preserve raw pixels and are used as input features in BEiT.

In our experiments, we split each $224 \times 224$ image into a $14 \times 14$ grid of image patches, where each patch is $16 \times 16$.

### 2.1.2 VISUAL TOKEN

Similar to natural language, we represent the image as a sequence of discrete tokens obtained by an "image tokenizer", instead of raw pixels. Specifically, we tokenize the image $x \in \mathbb{R}^{H \times W \times C}$ into $z = [z_1, \ldots, z_N] \in \mathcal{V}^{h \times w}$, where the vocabulary $\mathcal{V} = \{1, \ldots, |\mathcal{V}|\}$ contains discrete token indices.

Following (Ramesh et al., 2021), we use the image tokenizer learned by discrete variational autoencoder (dVAE). There are two modules during visual token learning, namely, *tokenizer* and *decoder*. The tokenizer $q_\phi(z|x)$ maps image pixels $x$ into discrete tokens $z$ according to a visual codebook (i.e., vocabulary). The decoder $p_\psi(x|z)$ learns to reconstruct the input image $x$ based on the visual tokens $z$. The reconstruction objective can be written as $\mathbb{E}_{z \sim q_\phi(z|x)}[\log p_\psi(x|z)]$. Because the latent visual tokens are discrete, the model training is non-differentiable. Gumbel-softmax relaxation (Jang et al., 2017; Maddison et al., 2017) is employed to train the model parameters. Moreover, a uniform prior is put on $q_\phi$ during dVAE training. Refer to (Ramesh et al., 2021) for more training details of the image tokenizer.

We tokenize each image to a $14 \times 14$ grid of visual tokens. Notice the number of visual tokens and the number of image patches for one image are the same. The vocabulary size is set to $|\mathcal{V}| = 8192$. In our work, we directly use the publicly available[1] image tokenizer described in (Ramesh et al., 2021). We also compare it with a re-implemented tokenizer in Appendix C.

## 2.2 BACKBONE NETWORK: IMAGE TRANSFORMER

Following ViT (Dosovitskiy et al., 2020), we use the standard Transformer (Vaswani et al., 2017) as the backbone network. So the results can be directly compared with previous work in terms of the network architecture.

The input of Transformer is a sequence of image patches $\{x_i^p\}_{i=1}^N$. The patches are then linearly projected to obtain patch embeddings $Ex_i^p$, where $E \in \mathbb{R}^{(P^2 C) \times D}$. Moreover, we prepend a special token [S] to the input sequence. We also add standard learnable 1D position embeddings $E_{pos} \in \mathbb{R}^{N \times D}$ to patch embeddings. The input vectors $H_0 = [e_{[S]}, Ex_i^p, \ldots, Ex_N^p] + E_{pos}$ is fed into Transformer. The encoder contains $L$ layers of Transformer blocks $H^l = \text{Transformer}(H^{l-1})$, where $l = 1, \ldots, L$. The output vectors of the last layer $H^L = [h_{[S]}^L, h_1^L, \ldots, h_N^L]$ are used as the encoded representations for the image patches, where $h_i^L$ is the vector of the $i$-th image patch.

---

[1]https://github.com/openai/DALL-E

## 2.3 Pre-Training BEiT: Masked Image Modeling

We propose a *masked image modeling* (MIM) task. We randomly mask some percentage of image patches, and then predict the visual tokens that are corresponding to the masked patches.

Figure 1 shows the overview of our method. As presented in Section 2.1, given an input image $x$, we split it into $N$ image patches ($\{x_i^p\}_{i=1}^N$), and tokenize it to $N$ visual tokens ($\{z_i\}_{i=1}^N$). We randomly mask approximately $40\%$ image patches, where the masked positions are denoted as $\mathcal{M} \in \{1, \ldots, N\}^{0.4N}$. Next we replace the masked patches with a learnable embedding $e_{[M]} \in \mathbb{R}^D$. The corrupted image patches $x^{\mathcal{M}} = \{x_i^p : i \notin \mathcal{M}\}_{i=1}^N \bigcup \{e_{[M]} : i \in \mathcal{M}\}_{i=1}^N$ are then fed into the $L$-layer Transformer as described in Section 2.2. The final hidden vectors $\{h_i^L\}_{i=1}^N$ are regarded as encoded representations of the input patches. For each masked position $\{h_i^L : i \in \mathcal{M}\}_{i=1}^N$, we use a softmax classifier to predict the corresponding visual tokens $p_{\text{MIM}}(z'|x^{\mathcal{M}}) = \text{softmax}_{z'}(W_c h_i^L + b_c)$, where $x^{\mathcal{M}}$ is the corrupted image, $W_c \in \mathbb{R}^{|\mathcal{V}| \times D}$, and $b_c \in \mathbb{R}^{|\mathcal{V}|}$. The pre-training objective is to maximize the log-likelihood of the correct visual tokens $z_i$ given the corrupted image:

$$\max \sum_{x \in \mathcal{D}} \mathbb{E}_{\mathcal{M}} \left[ \sum_{i \in \mathcal{M}} \log p_{\text{MIM}}(z_i|x^{\mathcal{M}}) \right] \tag{1}$$

where $\mathcal{D}$ is the training corpus, $\mathcal{M}$ represents randomly masked positions, and $x^{\mathcal{M}}$ is the corrupted image that is masked according to $\mathcal{M}$.

Rather than randomly choosing patches for the masked positions $\mathcal{M}$, we employ blockwise masking in our work. As summarized in Algorithm 1, a block of image patches is masked each time. For each block, we set the minimum number of patches to 16. Then we randomly choose an aspect ratio for the masking block. We repeat the above two steps until obtaining enough masked patches, i.e., $0.4N$, where $N$ is the total number of image patches, and $0.4$ is masking ratio.

---

**Algorithm 1** Blockwise Masking

**Input:** $N(= h \times w)$ image patches
**Output:** Masked positions $\mathcal{M}$
$\mathcal{M} \leftarrow \{\}$
**repeat**
$\quad s \leftarrow \text{Rand}(16, 0.4N - |\mathcal{M}|)$ ▷ *Block size*
$\quad r \leftarrow \text{Rand}(0.3, \frac{1}{0.3})$ ▷ *Aspect ratio of block*
$\quad a \leftarrow \sqrt{s \cdot r}; b \leftarrow \sqrt{s/r}$
$\quad t \leftarrow \text{Rand}(0, h - a) \, ; l \leftarrow \text{Rand}(0, w - b)$
$\quad \mathcal{M} \leftarrow \mathcal{M} \bigcup \{(i, j) : i \in [t, t + a), j \in [l, l + b)\}$
**until** $|\mathcal{M}| > 0.4N$ ▷ *Masking ratio is 40%*
**return** $\mathcal{M}$

---

The MIM task is greatly inspired by masked language modeling (Devlin et al., 2019), which is one of the most successful pre-training objective in natural language processing. Moreover, blockwise (or n-gram) masking is also widely applied in BERT-like models (Joshi et al., 2020; Bao et al., 2020; Raffel et al., 2020). However, directly using pixel-level auto-encoding (i.e., recovering the pixels of masked patches) for vision pre-training pushes the model to focus on short-range dependencies and high-frequency details (Ramesh et al., 2021). BEiT overcomes the above issue by predicting discrete visual tokens, which summarizes the details to high-level abstractions. Ablation studies in Section 3.3 show that our proposed method significantly outperforms pixel-level auto-encoding.

## 2.4 From the Perspective of Variational Autoencoder

The BEiT pre-training can be viewed as variational autoencoder (Kingma & Welling, 2014) training. Let $x$ denote the original image, $\tilde{x}$ the masked image, and $z$ the visual tokens. Considering the evidence lower bound (ELBO) of the log-likelihood $p(x|\tilde{x})$, i.e., recovering the original image from its corrupted version:

$$\sum_{(x_i, \tilde{x}_i) \in \mathcal{D}} \log p(x_i|\tilde{x}_i) \geq \sum_{(x_i, \tilde{x}_i) \in \mathcal{D}} \Big( \underbrace{\mathbb{E}_{z_i \sim q_\phi(z|x_i)}[\log p_\psi(x_i|z_i)]}_{\text{Visual Token Reconstruction}} - D_{\text{KL}}[q_\phi(z|x_i), p_\theta(z|\tilde{x}_i)] \Big) \tag{2}$$

where (1) $q_\phi(z|x)$ denotes the image tokenizer that obtains visual tokens; (2) $p_\psi(x|z)$ decodes the original image given input visual tokens; (3) $p_\theta(z|\tilde{x})$ recovers the visual tokens based on the masked image, which is our MIM pre-training task.

We learn the model following a two-stage procedure similar to (van den Oord et al., 2017; Razavi et al., 2019). In the first stage, we obtain the image tokenizer as a discrete variational autoencoder (Ramesh et al., 2021). Specifically, the first stage minimizes the reconstruction loss

$-\mathbb{E}_{z_i \sim q_\phi(\mathbf{z}|x_i)}[\log p_\psi(x_i|z_i)]$ with an uniform prior as described in Equation (2). In the second stage, we learn the prior $p_\theta$ while keeping $q_\phi$ and $p_\psi$ fixed. We simplify $q_\phi(\mathbf{z}|x_i)$ to a one-point distribution with the most likely visual tokens $\hat{z}_i = \arg\max_z q_\phi(z|x_i)$. Then Equation (2) can be rewritten as:

$$\sum_{(x_i,\tilde{x}_i) \in \mathcal{D}} \Big( \underbrace{\mathbb{E}_{z_i \sim q_\phi(z|x_i)}[\log p_\psi(x_i|z_i)]}_{\text{Stage 1: Visual Token Reconstruction}} + \underbrace{\log p_\theta(\hat{z}_i|\tilde{x}_i)}_{\text{Stage 2: Masked Image Modeling}} \Big) \qquad (3)$$

where the second term is our BEIT pre-training objective.

## 2.5 PRE-TRAINING SETUP

The network architecture of BEIT follows that of ViT-Base (Dosovitskiy et al., 2020) for a fair comparison. We use a 12-layer Transformer with 768 hidden size, and 12 attention heads. The intermediate size of feed-forward networks is 3072. We employ the default $16 \times 16$ input patch size. We directly borrow the image tokenizer trained by Ramesh et al. (2021). The vocabulary size of visual tokens is 8192.

We pretrain BEIT on the training set of ImageNet-1K (Russakovsky et al., 2015), which contains about 1.2M images. Our augmentation policy includes random resized cropping, horizontal flipping, color jittering (Wu et al., 2018). Notice that we do not use the labels for self-supervised learning. We use the $224 \times 224$ resolution in our experiments. So the input is split to $14 \times 14$ image patches, and the same amount of visual tokens. We randomly mask at most 75 patches (i.e., roughly $40\%$ of total image patches).

The pre-training runs for about 500k steps (i.e., 800 epochs) with 2k batch size. Adam (Loshchilov & Hutter, 2019) with $\beta_1 = 0.9, \beta_2 = 0.999$ is employed for optimization. The learning rate is set to 1.5e-3, with a warmup of 10 epochs, and cosine learning rate decay. The weight decay is 0.05. We employ stochastic depth (Huang et al., 2016) with a 0.1 rate, and disable dropout. The 500k training steps take about five days using 16 Nvidia Telsa V100 32GB GPU cards.

We find that proper initialization is important to stabilize Transformer, especially for large-scale pre-training. We first randomly initialize all the parameters within a small range, such as $[-0.02, 0.02]$. Then, for the $l$-th Transformer layer, we rescale the output matrices (i.e., the last linear projection within each sub-layer) of the self-attention module and the feed-forward network by $\frac{1}{\sqrt{2l}}$.

## 2.6 FINE-TUNING BEIT ON DOWNSTREAM VISION TASKS

After pre-training BEIT, we append a task layer upon the Transformer, and fine-tune the parameters on downstream tasks, like BERT. We take image classification and semantic segmentation as examples in our work. It is straightforward to leverage the pre-training-then-fine-tuning paradigm on other vision tasks with BEIT.

**Image classification.** For image classification tasks, we directly employ a simple linear classifier as the task layer. Specifically, we use average pooling to aggregate the representations, and feed the global to a softmax classifier. The category probabilities are computed as $\text{softmax}(\text{avg}(\{\boldsymbol{h}_i^L\}_{i=1}^N \boldsymbol{W}_c))$, where $\boldsymbol{h}_i^L$ is the final encoding vector of the $i$-th image patch, $\boldsymbol{W}_c \in \mathbb{R}^{D \times C}$ is a parameter matrix, and $C$ is the number of labels. We maximize the likelihood of labeled data by updating the parameters of BEIT and the softmax classifier.

**Semantic segmentation.** For semantic segmentation, we follow the task layer used in SETR-PUP (Zheng et al., 2020). To be specific, we use pretrained BEIT as a backbone encoder, and incorporate several deconvolution layers as decoder to produce segmentation. The model is also end-to-end fine-tuned similar to image classification.

**Intermediate fine-tuning.** After self-supervised pre-training, we can further train BEIT on a data-rich intermediate dataset (i.e., ImageNet-1K in our work), and then finetune the model on the target downstream tasks. Such intermediate fine-tuning is the common practice of BERT fine-tuning in NLP (Pruksachatkun et al., 2020). We directly follow the method for BEIT.

## 3 EXPERIMENTS

We conduct full fine-tuning experiments on image classification and semantic segmentation. Moreover, we present various ablation studies for pre-training and analyze the representations learned by BEIT. We also report linear probes on ImageNet in Appendix D.

### 3.1 IMAGE CLASSIFICATION

The image classification task classifies input images to various categories. We evaluate BEIT on the ILSVRC-2012 ImageNet dataset (Russakovsky et al., 2015) with 1k classes and 1.3M images. We directly follow the most of hyperparameters of DeiT (Touvron et al., 2020) in our fine-tuning experiments for a fair comparison. We reduce fine-tuning epochs compared with training from scratch, as BEIT has been pre-trained. Accordingly, we use a larger learning rate with layer-wise decay. The detailed hyperparameters are summarized in Appendix H.

Table 1 reports top-1 accuracy on image classification. We compare BEIT with vision Transformers trained by random initialization, supervised pre-training, and previous self-supervised learning methods. All the compared models are base-size, except iGPT has 1.36B parameters. Pre-training is conducted on ImageNet for the comparison purpose, except ViT-JFT300M is pretrained on Google's in-house 300M images.

Compared with the models trained by random initialization, we find that pre-trained BEIT significantly improves performance on both datasets. BEIT improves the performance on ImageNet, which shows the effectiveness under the rich-resource setting.

Moreover, we compare BEIT with previous state-of-the-art self-supervised methods for Transformer, such as DINO (Caron et al., 2021), and MoCo v3 (Chen et al., 2021). Our proposed method outperforms previous models on ImageNet fine-tuning. Among them, iGPT-1.36B (Chen et al., 2020a) uses much more parameters (i.e., 1.36B vs 86M), and ViT-JFT300M (Dosovitskiy et al., 2020) is pretrained on larger corpus (i.e., 300M vs 1.3M), while others pretrain ViT-Base on ImageNet-1K. iGPT-1.36B and ViT-JFT300M are the most comparable methods, which also follows auto-encoding pre-training for vision Transformer. Specifically, iGPT uses clustered image tokens as both input and output for image GPT or image BERT. In contrast, we use image patches as input to preserve raw pixels, and employ discrete visual tokens as a prediction bottleneck. ViT-JFT300 predicts the mean, 3-bit color of each masked patch, rather than visual tokens learned by discrete VAE. We also pretrain the self-supervised tasks of BEIT and DINO in a multi-task learning manner, which is presented in Appendix E.

In addition, we evaluate our proposed method with intermediate fine-tuning. In other words, we first pretrain BEIT in a self-supervised manner, and then fine-tune the pretrained model on ImageNet with labeled data. The results show that BEIT is complementary to supervised pre-training, achieving additional gain after intermediate fine-tuning on ImageNet.

**Fine-tuning to $384 \times 384$ resolution.** After fine-tuning with resolution $224 \times 224$, we additionally fine-tune the model on $384 \times 384$ images by 10 more epochs. We follow the standard higher-resolution setting of DeiT (Touvron et al., 2020), except using fewer epochs. Notice that we keep patch size the same for both $224 \times 224$ and $384 \times 384$ images. So the input sequence length of Transformers becomes longer for higher resolutions. Table 1 shows that higher resolution improves the BEIT results by 1+ points on ImageNet. More importantly, $\text{BEIT}_{384}$ pretrained on ImageNet-1K even outperforms supervised pre-training $\text{ViT}_{384}$ that uses ImageNet-22K, when they use the same input resolution.

**Scaling up to larger size.** We further scale up BEIT to the large size (same as ViT-L). As shown in Table 1, $\text{ViT}_{384}$-L is worse than $\text{ViT}_{384}$ on ImageNet, when training from scratch. The results verifies the data-hungry issue of vision Transformers. Supervised pre-training on ImageNet-22K partially relieves the issue, where $\text{ViT}_{384}$-L finally outperforms $\text{ViT}_{384}$ by 1.2. In comparison, BEIT-L is better than BEIT by 2.0, and $\text{BEIT}_{384}$-L outperforms $\text{BEIT}_{384}$ by 1.7. In other words, the benefits of scaling up BEIT from base to large are greater than supervised pre-training with ImageNet-22K. More importantly, comparing between $\text{BEIT}_{384}$ with $\text{ViT}_{384}$ that conducts supervised pre-training on ImageNet-22K, the improvements of BEIT become greater along with scaling the size from base

| Models | Model Size | Resolution | ImageNet |
|---|---|---|---|
| *Training from scratch (i.e., random initialization)* | | | |
| ViT$_{384}$-B (Dosovitskiy et al., 2020) | 86M | $384^2$ | 77.9 |
| ViT$_{384}$-L (Dosovitskiy et al., 2020) | 307M | $384^2$ | 76.5 |
| DeiT-B (Touvron et al., 2020) | 86M | $224^2$ | 81.8 |
| DeiT$_{384}$-B (Touvron et al., 2020) | 86M | $384^2$ | 83.1 |
| *Supervised Pre-Training on ImageNet-22K (using labeled data)* | | | |
| ViT$_{384}$-B (Dosovitskiy et al., 2020) | 86M | $384^2$ | 84.0 |
| ViT$_{384}$-L (Dosovitskiy et al., 2020) | 307M | $384^2$ | 85.2 |
| *Self-Supervised Pre-Training on ImageNet-1K (without labeled data)* | | | |
| iGPT-1.36B[†] (Chen et al., 2020a) | 1.36B | $224^2$ | 66.5 |
| ViT$_{384}$-B-JFT300M[‡] (Dosovitskiy et al., 2020) | 86M | $384^2$ | 79.9 |
| MoCo v3-B (Chen et al., 2021) | 86M | $224^2$ | 83.2 |
| MoCo v3-L (Chen et al., 2021) | 307M | $224^2$ | 84.1 |
| DINO-B (Caron et al., 2021) | 86M | $224^2$ | 82.8 |
| BEIT-B (ours) | 86M | $224^2$ | 83.2 |
| BEIT$_{384}$-B (ours) | 86M | $384^2$ | 84.6 |
| BEIT-L (ours) | 307M | $224^2$ | 85.2 |
| BEIT$_{384}$-L (ours) | 307M | $384^2$ | **86.3** |

Table 1: Top-1 accuracy on ImageNet-1K. We evaluate base- ("-B") and large-size ("-L") models at resolutions $224 \times 224$ and $384 \times 384$. [†]: iGPT-1.36B contains 1.36 billion parameters, while others are base-size models. [‡]: ViT$_{384}$-B-JFT300M is pretrained with the "masked patch prediction" task on Google's in-house 300M images, while others use ImageNet.

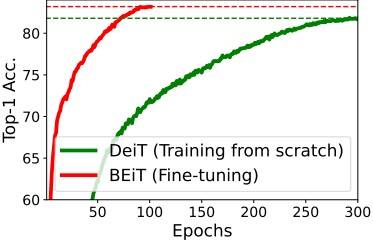

Table 2: Convergence curves of training DeiT from scratch and fine-tuning BEIT on ImageNet-1K.

| Models | ADE20K |
|---|---|
| Supervised Pre-Training on ImageNet | 45.3 |
| DINO (Caron et al., 2021) | 44.1 |
| BEIT (ours) | 45.6 |
| BEIT + Intermediate Fine-Tuning (ours) | **47.7** |

Table 3: Results of semantic segmentation on ADE20K. We use SETR-PUP (Zheng et al., 2020) as the task layer and report results of single-scale inference.

(i.e., 0.6) to large (i.e., 1.1). The results suggest that BEIT tends to help more for extremely larger models (such as 1B, or 10B), especially when labeled data are insufficient[2] to conduct supervised pre-training[3] for such large models.

**Convergence curves.** Figure 2 compares the convergence curves of the training-from-scratch and pre-training-then-fine-tuning paradigms. We find that fine-tuning BEIT not only achieves better performance, but also converging much faster than training DeiT from scratch. Moreover, fine-tuning BEIT can reach reasonable numbers within very few epochs.

---

[2]Zhai et al. (2021) report that supervised pre-training of a 1.8B-size vision Transformer requires billions of labeled images.

[3]Appendix B shows that BEIT fine-tuned on ImageNet-22K (14M) can match the performance of supervised pre-training on Google's in-house JFT-3B (Zhai et al., 2021), while using 214x less labels. We also demonstrate that large-size BEIT fine-tuned on 70M labeled images can achieve $89.5\%$ top-1 accuracy on ImageNet and $58.4\%$ mIoU on ADE20K, creating new state-of-the-art results for large-size vision Transformers.

| Models | ImageNet | ADE20K |
|---|---|---|
| BEiT (300 Epochs) | 82.86 | 44.65 |
| − Blockwise masking | 82.77 | 42.93 |
| − Visual tokens (i.e., recover masked pixels) | 81.04 | 41.38 |
| − Visual tokens − Blockwise masking | 80.50 | 37.09 |
| + Recover 100% visual tokens | 82.59 | 40.93 |
| − Masking + Recover 100% visual tokens | 81.67 | 36.73 |
| Pretrain longer (800 epochs) | 83.19 | 45.58 |

Table 4: Ablation studies for BEiT pre-training on image classification and semantic segmentation.

## 3.2 Semantic Segmentation

Semantic segmentation aims to predict a corresponding class for each pixel of the input image. We evaluate BEiT on the ADE20K benchmark (Zhou et al., 2019) with 25K images and 150 semantic categories. We report the metric of mean Intersection of Union (mIoU) averaged over all semantic categories. As presented in Section 2.6, we directly follow the task layer and the most of hyperparameters described in SETR-PUP (Zheng et al., 2020). On ADE20K, we use Adam (Loshchilov & Hutter, 2019) as the optimizer. The learning rate is set to 1e-3 with layer-wise decay similar to image classification. We conduct fine-tuning for 160K steps. The batch size is 16. The detailed hyperparameters are described in Appendix I.

As shown in Table 3, we compare BEiT with supervised pre-training that relies on labeled data of ImageNet. We find that our proposed method achieves better performance than supervised pre-training, although BEiT does not require manual annotations for pre-training. Moreover, we employ intermediate fine-tuning for BEiT on ImageNet, i.e., we first fine-tune pretrained BEiT on ImageNet, and then fine-tune the model on ADE20K. The results indicate that intermediate fine-tuning further improves BEiT on semantic segmentation.

## 3.3 Ablation Studies

We conduct ablation studies to analyze the contributions of each component in BEiT. The models are evaluated on image classification (i.e., ImageNet) and semantic segmentation (i.e., ADE20K). We set the default pre-training steps to 300 epochs for the ablation studies, which is 37.5% of the total steps used in the previous experiments.

Table 4 reports the results of various model variants. First, we ablate blockwise masking by randomly sample masked positions. We find that blockwise masking is beneficial on both tasks, especially on semantic segmentation. Second, we ablate the usage of visual tokens by predicting the raw pixels of masked patches, i.e., the pre-training task becomes a pixel regression problem to recover masked patches. Our proposed masked image modeling task significantly outperforms naive pixel-level auto-encoding. Compared with the results in Table 1, the ablation result is worse than training vision Transformer from scratch on two tasks. The results indicate that the prediction of visual tokens is the key ingredient of BEiT. Third, we ablate the usage of visual tokens and blockwise masking together. We find that blockwise masking is even more helpful for pixel-level auto-encoding, which relieves the suffering of short-distance dependency. Forth, recovering all the visual tokens harms performance on downstream tasks. Fifth, we compare BEiT with different training steps. Pre-training the model longer can further improve performance on downstream tasks.

## 3.4 Analysis of Self-Attention Map

We show that the self-attention mechanism in BEiT can separate objects, even though our pre-training does not rely on any manual annotation at all. Similar properties are also observed by Caron et al. (2021). The probing images are taken from the MS COCO (Lin et al., 2014) corpus to avoid appearing in the pre-training data.

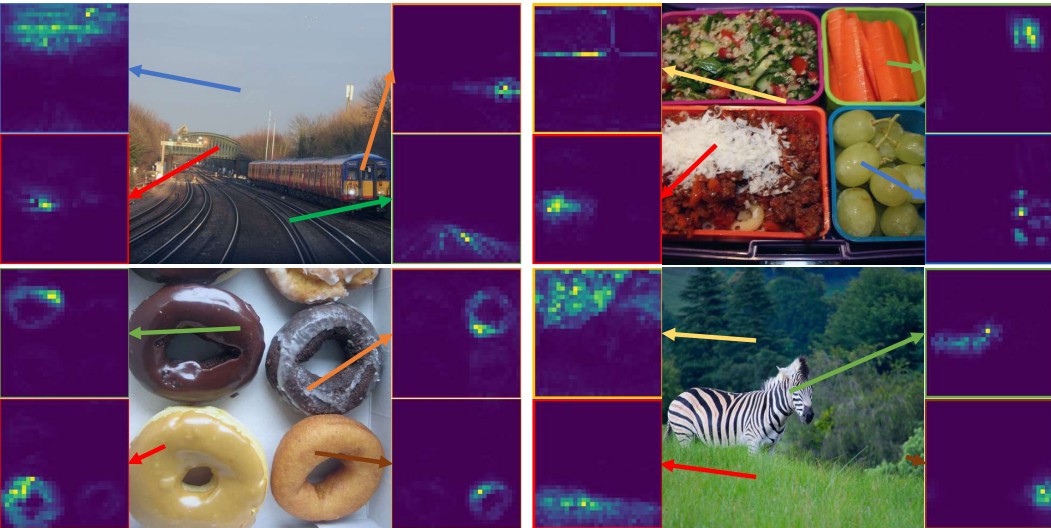

Figure 2: Self-attention map for different reference points. The self-attention mechanism in BEIT is able to separate objects, although self-supervised pre-training does not use manual annotations.

As shown in Figure 2, we plot the self-attention map for different reference points within an image. The visualizations are produced by attention scores computed via query-key product in the last layer. For each reference point, we use the corresponding patch as query, and show which patch it attends to. After pre-training, BEIT learns to distinguish semantic regions using self-attention heads, without any task-specific supervision. The property partially indicates the reason why BEIT is able to help downstream tasks. Such knowledge acquired by BEIT potentially improves the generalization ability of fine-tuned models, especially on small-scale datasets.

## 4 RELATED WORK

**Self-supervised visual representation learning.** Various methods have been introduced over the years to pretrain vision models in a self-supervised manner. Pioneering works design clever pretext tasks, such as predicting the patch orderings (Noroozi & Favaro, 2016), colorization (Zhang et al., 2016), and predicting rotation angles (Komodakis & Gidaris, 2018). In addition, Trinh et al. (2019) propose to mask some patches within an image, and classify whether the masked patches are real or fake for each masked position. The method is similar to the masked version of Jigsaw pre-training (Noroozi & Favaro, 2016). The recent strand of research follows contrastive paradigm (Wu et al., 2018; Oord et al., 2018; Hjelm et al., 2019; Bachman et al., 2019; He et al., 2020; Chen et al., 2020b;c). The models typically regard various data augmentations as different views of an image, and then make the representations of positive pairs similar while pushing negative pairs away. In order to obtain enough informative negative samples in contrastive learning, the methods usually rely on large memory banks (Wu et al., 2018; He et al., 2020) or large batch size (Chen et al., 2020b). BYOL (Grill et al., 2020) and SimSiam (Chen & He, 2020) further eliminate the requirement of negative samples, using various techniques to avoid representation collapse. Another strand of methods use clustering to organize image examples (Caron et al., 2018; Asano et al., 2020; Caron et al., 2020; Li et al., 2021).

**Self-supervised vision Transformers.** Pre-training vision Transformers has received significant attention recently due to the data-hungry issue. iGPT (Chen et al., 2020a) first creates a 9-bit color palette by k-means clustering RGB pixels, and then uses the clustered tokens to represent images. Next iGPT uses the tasks of BERT and GPT to pretrain Transformers. In comparison, our proposed method uses image patches as input without losing pixel-level information. Moreover, our visual tokens are obtained by discrete VAE instead of clustering. ViT (Dosovitskiy et al., 2020) conducts a preliminary exploration with the masked patch prediction task, which predicts the 3-bit mean color of the masked patches. Dosovitskiy et al. (2020) also report that pixel-level auto-encoding performs

worse, although it is the most straightforward translation of BERT from NLP to CV. Rather than using heuristically designed pre-training tasks, our proposed model leverages visual tokens learned by discrete VAE, which not only achieves better performance but also is better theoretically motivated. Apart from masked auto-encoding, other mainstream research works use contrastive learning (Chen et al., 2021; Xie et al., 2021), and self-distillation (Caron et al., 2021). In comparison, BEIT can achieve several times of improvement in terms of pre-training throughput (Appendix E), and memory consumption. The advantages make BEIT appealing to scale up vision Transformers.

## 5 CONCLUSION

We introduce a self-supervised pre-training framework for vision Transformers, achieving strong fine-tuning results on downstream tasks, such as image classification, and semantic segmentation. We show that the proposed method is critical to make BERT-like pre-training (i.e., auto-encoding with masked input) work well for image Transformers. We also present the intriguing property of automatically acquired knowledge about semantic regions, without using any human-annotated data. In the future, we would like to scale up BEIT pre-training in terms of data size and model size. Moreover, we will conduct multimodal pre-training in a more unified way, using the similar objectives and the shared architecture for texts and images.

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

## A   ARCHITECTURE VARIANTS OF VISION TRANSFORMER

We use the standard vision Transformer (ViT; Dosovitskiy et al. 2020) in the experiments for fair comparisons. In addition, we find that LayerScale (Touvron et al., 2021) and relative position bias (Bao et al., 2020; Raffel et al., 2020) improve ViTs on downstream tasks. We employ the same setting as in Section 3.3 for ablation studies, which pretrains base-size models for 300 epochs on ImageNet-1K.

As shown in Table 5, both LayerScale and relative position bias improve performance on ImageNet classification and ADE20K semantic segmentation. We denote the improved architecture as BEIT$^+$ and use it for the experiments in Appendix B. We empirically notice that vanilla Transformer is the most stable when scaling up the model to billions of parameters, so we do not use LayerScale for extra-large models.

| Architecture | ImageNet | ADE20K |
|---|---|---|
| ViT (used in this paper) | 82.86 | 44.86 |
| ViT+LayerScale | 83.00 | 45.43 |
| ViT+LayerScale+Relative Position Bias | 83.22 | 45.70 |

Table 5: Ablation studies of architecture variants on image classification and semantic segmentation. For ADE20K, we use UperNet (Xiao et al., 2018) as the task layer, and report mIoU scores of single-scale inference.

## B   COMPARISON WITH LARGE-SCALE SUPERVISED PRE-TRAINING

We compare with state-of-the-art supervised pre-training at scale. In addition to using ImageNet-1K for fair comparisons with previous work, we pretrain BEIT on ImageNet-22K to boost performance. We employ the architecture improvements (i.e., LayerScale, and relative position bias) as described in Appendix A, which is denoted as BEIT$^+$ in Table 6 and Table 7. We follow the same pre-training setup as in Section 2.5, except we pretrain 150 epochs on ImageNet-22K. After self-supervised pre-training, we conduct intermediate fine-tuning on ImageNet-22K for 90 epochs. Moreover, we use an in-house dataset that has about 70M labeled images as a drop-in replacement of ImageNet-22K.

| Models | Model Size | Labeled Data Size | ImageNet | |
|---|---|---|---|---|
| | | | $384^2$ | $512^2$ |
| *Supervised Pre-Training on ImageNet-22K (using labeled data)* | | | | |
| ViT-B (Dosovitskiy et al., 2020) | 86M | 14M | 84.0 | - |
| ViT-L (Dosovitskiy et al., 2020) | 307M | 14M | 85.2 | 85.30 |
| ViT-H (Dosovitskiy et al., 2020) | 632M | 14M | 85.1 | - |
| *Supervised Pre-Training on Google JFT-300M (using labeled data)* | | | | |
| ViT-B (Dosovitskiy et al., 2020) | 86M | 300M | 84.2 | - |
| ViT-L (Dosovitskiy et al., 2020) | 307M | 300M | 87.1 | 87.76 |
| ViT-H (Dosovitskiy et al., 2020) | 632M | 300M | 88.0 | 88.55 |
| *Supervised Pre-Training on Google JFT-3B (using labeled data)* | | | | |
| ViT-B (Zhai et al., 2021) | 86M | 3000M | 86.6 | - |
| ViT-L (Zhai et al., 2021) | 307M | 3000M | 88.5 | - |
| *Self-Supervised Pre-Training, and Intermediate Fine-Tuning on ImageNet-22K* | | | | |
| BEIT-B$^+$ (ours) | 86M | 14M | 86.8 | - |
| BEIT-L$^+$ (ours) | 307M | 14M | 88.4 | 88.6 |
| *Self-Supervised Pre-Training, and Intermediate Fine-Tuning on In-House-70M* | | | | |
| BEIT-L$^+$ (ours) | 307M | 70M | **89.3** | **89.5** |

Table 6: Top-1 accuracy on ImageNet-1K fine-tuning. We evaluate models at resolutions $384^2$ and $512^2$.

Table 6 compares BEIT with previous state-of-the-art supervised pre-training (Dosovitskiy et al., 2020; Zhai et al., 2021) on ImageNet fine-tuning. Rather than heavily relying on extremely large-size labeled data (such as Google's in-house JFT-300M and JFT-3B), we demonstrate that BEIT pre-training can catch up with only ImageNet-22k (14M). Specifically, BEIT-L fine-tuned on ImageNet-22K achieves comparable performance with ViT-L trained on Google JFT-3B. Moreover, BEIT-L obtains $89.5\%$ top-1 accuracy on ImageNet after intermediate fine-tuning on an in-house 70M dataset. The results indicate that BEIT pre-training greatly reduces the required labeling efforts and advances the new state of the art for large-size vision Transformers.

As shown in Table 7, we report the fine-tuning results on the ADE20K semantic segmentation benchmark. Following Swin (Liu et al., 2021b), we use the same task layer (i.e., UperNet; Xiao et al. 2018) and evaluate the models at the resolution $640 \times 640$. The BEIT-L model obtains state-of-the-art performance on ADE20K.

| Models | mIoU (%) | Multi-Scale mIoU (%) |
|---|---|---|
| *Supervised Pre-Training on ImageNet-22K (using labeled data)* | | |
| Swin-B (Liu et al., 2021b) | 50.0 | 51.7 |
| Swin-L (Liu et al., 2021b) | 52.1 | 53.5 |
| *Self-Supervised Pre-Training, and Intermediate Fine-Tuning on ImageNet-22K* | | |
| BEIT-B$^+$ (ours) | 53.6 | 54.2 |
| BEIT-L$^+$ (ours) | 56.7 | 57.0 |
| *Self-Supervised Pre-Training, and Intermediate Fine-Tuning on In-House-70M* | | |
| BEIT-L$^+$ (ours) | **57.9** | **58.4** |

Table 7: Performance comparison on the ADE20K semantic segmentation. We follow Swin-L (Liu et al., 2021b) to use UperNet (Xiao et al., 2018) as the task layer and evaluate at resolution $640 \times 640$.

## C  ABLATION STUDIES OF IMAGE TOKENIZER

For comparison, we re-train the image tokenizer on ImageNet-1K. The reimplementation is based on `https://github.com/lucidrains/DALLE-pytorch`. We use the same codebook size 8K as in DALL-E (Ramesh et al., 2021). Then we plug the tokenizer into our pre-training process. We follow the same experimental setup of ablation studies as in Section 3.3. Table 8 shows that our reimplemented tokenizer obtains comparable reconstruction loss and ImageNet fine-tuning performance compared with the off-the-shelf DALL-E tokenizer.

| Image Tokenizer | Reconstruction Error | ImageNet |
|---|---|---|
| DALL-E Tokenizer (Ramesh et al., 2021) | **0.0856** | **82.86** |
| Our reimplementation | 0.0880 | 82.70 |

Table 8: Top-1 accuracy on ImageNet-1K using different image tokenizers during pre-training. For image reconstruction, we report mean absolute error of normalized RGB values. The reimplemented image tokenizer is trained on ImageNet-1K without labels.

## D  LINEAR PROBES ON IMAGENET

We evaluate linear probes on ImageNet for various pretrained vision Transformers. We compare BEIT with two main strands of work, namely *discriminative* and *generative* self-supervised learning. The first one applies discriminative learning for pre-training, such as contrastive learning (Chen et al., 2021), and self distillation (Caron et al., 2021). The above methods typically learn to aggregate the image-level features into a global vector, which is relatively suitable for linear probing. In contrast, the second strand of methods, such as iGPT (Chen et al., 2020a) and ours, usually do not pretrain such global feature aggregation, which tends to make linear probes difficult.

Following iGPT (Chen et al., 2020a), we use average pooling to aggregate the hidden states of each image patches, and add the probing layer at the middle layer of Transformer instead of always at the final layer. Similarly, we find that the best layer lies in 9-th layer for BEіT-B, and 14-th layer for BEіT-L. To be specific, we use AdamW (Loshchilov & Hutter, 2019) to update the linear probe layer for 50 epochs. The learning rate is 4e-3 with cosine decay. The batch size is 1024. The weight decay is set to 1e-4. We follow data augmentation used in DINO (Caron et al., 2021), which uses random resize crops and horizontal flips augmentation during training and evaluates on central crops.

| Models | Model Size | Accuracy |
|---|---|---|
| *Discriminative self-supervised learning* | | |
| DINO-B (Caron et al., 2021) | 86M | 78.2 |
| MoCo v3-B (Chen et al., 2021) | 86M | 76.7 |
| MoCo v3-L (Chen et al., 2021) | 307M | 77.6 |
| *Generative self-supervised learning* | | |
| iGPT-L (Chen et al., 2020a) | 1362M | 65.2 |
| iGPT-XL (Chen et al., 2020a) | 6801M | 68.7 |
| iGPT-XL (Chen et al., 2020a) | 6801M | 72.0* |
| BEіT-B (ours) | 86M | 56.7 |
| BEіT-L (ours) | 307M | 73.5 |

Table 9: Linear probing accuracy on ImageNet. "∗" denotes that iGPT-XL uses concatenation of five layers for linear probing, while others use the features of single layer.

As shown in Table 9, we evaluate linear probes on ImageNet-1K for self-supervised learning. Overall, discriminative methods perform better than generative pre-training on linear probing. Linear probes keep the Transformer parameters fixed and only update the linear layer. So the pre-training of global aggregation of image-level features is beneficial to linear probing in DINO and MoCo v3, although full fine-tuning eliminates the gap. Moreover, the results indicate that increasing the model size from base (86M) to large (304M) significantly improves accuracy for our proposed method. In contrast, the gap between base- and large-size MoCo v3 is smaller. We also find that BEіT outperforms iGPT by a large margin even using much fewer parameters.

## E    MULTI-TASK PRE-TRAINING WITH DINO

We train the pre-training tasks of BEіT and DINO (Caron et al., 2021) together in a multi-task manner. As shown in Table 10, augmenting masked image modeling with DINO improves semantic segmentation on ADE20K, and obtains comparable results on ImageNet classification. Moreover, BEіT is more efficient in terms of pre-training speed, as DINO has two copies of Transformer parameters for self-distillation and multi-crop augmentation (Caron et al., 2020). For the throughput comparisons between BEіT and BEіT+DINO, we set batch size to the same. Because BEіT is also more memory-efficient, we can use larger batch size to fully utilize GPU cards, which obtains greater speedup in practice than the reported numbers.

| Models | ImageNet | ADE20K | Pre-Training Throughput |
|---|---|---|---|
| DINO (400 Epochs) | 82.8 | 44.08 | - |
| BEіT (300 Epochs) | **82.9** | 44.65 | **4.2x** |
| BEіT + DINO (300 Epochs) | **82.9** | **46.85** | 1.0x |

Table 10: We train the pre-training tasks of BEіT and DINO (Caron et al., 2021) in the way of multi-task learning. We report the performance by fine-tuning on ImageNet-1K image classification and ADE20K semantic segmentation. For ADE20K, we use SETR-PUP (Zheng et al., 2020) as the task layer and report the mIoU score of single-scale inference. The pre-training throughput measures the speed, where larger numbers indicate faster pre-training.

## F    IMAGE CLASSIFICATION ON CIFAR-100

In addition to ImageNet classification, we conduct fine-tuning experiments on the CIFAR-100 (Krizhevsky & Hinton, 2009) benchmark with 100 classes and 60k images. The experimental setup is the same as in Section 3.1.

Table 11 reports the top-1 accuracy on CIFAR-100. Notably, on the smaller CIFAR-100 dataset, ViT trained from scratch only reaches $48.5\%$ accuracy (Chen et al., 2021). In comparison, BEIT achieves $90.1\%$ with the help of pre-training. The results indicate that BEIT can greatly reduce the requirement of annotation efforts. BEIT also outperforms MoCo v3. Moreover, intermediate fine-tuning on ImageNet-1K further improves the results on CIFAR-100.

| Models | CIFAR-100 |
|---|:---:|
| *Training from scratch (i.e., random initialization)* | |
| ViT$_{384}$ (Dosovitskiy et al., 2020) | 48.5* |
| *Supervised Pre-Training on ImageNet-1K (using labeled data)* | |
| ViT$_{384}$ (Dosovitskiy et al., 2020) | 87.1 |
| DeiT (Touvron et al., 2020) | 90.8 |
| *Self-Supervised Pre-Training on ImageNet-1K (without labeled data)* | |
| DINO (Caron et al., 2021) | 91.7 |
| MoCo v3 (Chen et al., 2021) | 87.1 |
| BEIT (ours) | 90.1 |
| *Self-Supervised Pre-Training, and Intermediate Fine-Tuning on ImageNet-1K* | |
| BEIT (ours) | **91.8** |

Table 11: Top-1 accuracy of image classification on CIFAR-100. The models are at resolution $224 \times 224$, except ViT$_{384}$ uses $384 \times 384$. The results, unless otherwise indicated, are all obtained by base-size models. *: result is taken from (Chen et al., 2021).

## G    HYPERPARAMETERS FOR PRE-TRAINING

| Hyperparameters | Base Size | Large Size |
|---|:---:|:---:|
| Layers | 12 | 24 |
| Hidden size | 768 | 1024 |
| FFN inner hidden size | 3072 | 4096 |
| Attention heads | 12 | 16 |
| Attention head size | 64 | |
| Patch size | $16 \times 16$ | |
| Training epochs | 800 | |
| Batch size | 2048 | |
| Adam $\epsilon$ | 1e-8 | |
| Adam $\beta$ | (0.9, 0.999) | |
| Peak learning rate | 1.5e-3 | |
| Minimal learning rate | 1e-5 | |
| Learning rate schedule | Cosine | |
| Warmup epochs | 10 | |
| Gradient clipping | 3.0 | 1.0 |
| Dropout | ✗ | |
| Stoch. depth | 0.1 | |
| Weight decay | 0.05 | |
| Data Augment | RandomResizeAndCrop | |
| Input resolution | $224 \times 224$ | |
| Color jitter | 0.4 | |

Table 12: Hyperparameters for pre-training BEIT on ImageNet-1K.

## H  HYPERPARAMETERS FOR IMAGE CLASSIFICATION FINE-TUNING

| Hyperparameters | CIFAR-100 Base Size | ImageNet-1K Base Size | Large Size |
|---|---|---|---|
| Peak learning rate | {2e-3, 3e-3, 4e-3, 5e-3} | | |
| Fine-tuning epochs | 150 | 100 | 50 |
| Batch size | 512 | 1024 | 1024 |
| Warmup epochs | 20 | 20 | 5 |
| Layer-wise learning rate decay | 0.65 | 0.65 | 0.75 |
| Adam $\epsilon$ | 1e-8 | | |
| Adam $\beta$ | (0.9, 0.999) | | |
| Minimal learning rate | 1e-6 | | |
| Learning rate schedule | Cosine | | |
| Repeated Aug | ✓ | ✓ | ✗ |
| Weight decay | 0.3 | 0.05 | 0.05 |
| Label smoothing $\varepsilon$ | 0.1 | | |
| Stoch. depth | 0.1 | | |
| Dropout | ✗ | | |
| Gradient clipping | ✗ | | |
| Erasing prob. | ✗ | 0.25 | 0.25 |
| Input resolution | $224 \times 224$ | | |
| Rand Augment | 9/0.5 | | |
| Mixup prob. | 0.8 | | |
| Cutmix prob. | 1.0 | | |

Table 13: Hyperparameters for fine-tuning BEiT on ImageNet-1K and CIFAR-100.

## I  HYPERPARAMETERS FOR ADE20K SEMANTIC SEGMENTATION FINE-TUNING

| Hyperparameters | Base Size |
|---|---|
| Peak learning rate | 1e-3 |
| Fine-tuning steps | 160K |
| Batch size | 16 |
| Adam $\epsilon$ | 1e-8 |
| Adam $\beta$ | (0.9, 0.999) |
| Layer-wise learning rate decay | 0.65 |
| Minimal learning rate | 0 |
| Learning rate schedule | Linear |
| Warmup steps | 1500 |
| Dropout | ✗ |
| Stoch. depth | 0.1 |
| Weight decay | 0.05 |
| Input resolution | $512 \times 512$ |
| Position embedding interpolate | bilinear |

Table 14: Hyperparameters for fine-tuning BEiT on ADE20K.

