# OpenReview forum: "BEiT: BERT Pre-Training of Image Transformers"
_ICLR.cc/2022/Conference — ICLR 2022 Oral_

### Official Review · Reviewer_GYF1 · 2021-10-27

**Correctness:** 3
**Technical Novelty And Significance:** 3
**Empirical Novelty And Significance:** 3
**Recommendation:** 8
**Confidence:** 5

**Main Review:**

This paper presents a self-supervised learning framework named BEIT, in which the input image can be masked in some regions, and the task is to recover the token of the masked region. Pre-trained on ImageNet, the model shows good performance on a series of downstream tasks.

I shall say that I very much like the idea of this paper. Self-supervised learning in computer vision seems far behind that in natural language processing, mostly because the proxy task is not good enough. This is highly related to the original data format -- masked language modeling is a perfect proxy for texts, but it seems very difficult to replicate it on images -- because image signals are mostly continuous in the space, and semantic signals are sparse, etc. This paper makes a good trial along this direction. I think the mask image modeling task is much more elegant than the existing contrastive learning (e.g. SimCLR & MoCo) or predictive learning (e.g. BYOL) counterparts, because those methods are mostly relying on data augmentation to provide the prior-to-be-learned, but data augmentation not strong enough. More importantly, data augmentation can bring conflicts, e.g. one needs to enlarge the intensity of data augmentation to improve the difficulty of learning, but strong data augmentation is risky and may generate duals with very different semantics (so that they are bad for SSL).

OK, I have expressed my opinions that this new direction is promising. Also, the experiments on ImageNet and ADE20K are good (though I expect to see more including ImageNet linear-tests, MS-COCO tests, etc.), showing the strong ability of the pre-trained networks. However, I have one major concern that avoids me from giving a higher score to this paper. **That is, I am not sure if the proposed framework is fairly compared against the prior methods.**

The key lies in the tokenizer, which delivers almost all priors in the BEIT algorithm. However, the tokenizers are borrowed from DALL-E, which means that a large number of image-text pairs have been used -- the authors did not specify which model has been used, but at least, the CC3M (if not JFT300M) dataset is included in the pre-training part. This is not considered self-supervised learning, as the texts can contain vast amount of semantics (according to the CLIP paper, after training on image-text pairs, the model can achieve good performance on zero-shot ImageNet classification). I do not believe that using this tokenizer as the "teacher" to "distill" the target "student" model is "self-supervised learning". In Appendix C, the authors claimed that a re-implemented tokenizer on ImageNet shows similar performance, but technical details are missing. Even in the unsupervised setting, if the tokenizer is trained for sufficiently long, it can offer powerful guidance to the target network, so that some statements (e.g. Section 2.5, The pre-training runs for about 500k steps (i.e., 800 epochs)) are not strictly meaningful.

I hope the authors can answer the following questions.

1. The detailed setting of the DALL-E tokenizer and ImageNet re-trained tokenizer. In particular, please specify what kind of external information has been used, and the cost of pre-training such tokenizers.

2. If indeed external information is used (e.g. ImageNet labels to the worst case), is it possible to re-implement a counterpart without any such information and re-test the performance? Also, I am very interested in the performance of a random tokenizer (i.e. without pre-training, just clustering the responses of the randomly initialized networks).

By any means, I think a discussion on the relationship between this approach and knowledge distillation is necessary. I shall re-evaluate this paper after seeing answers to the above questions.

**Summary Of The Paper:**

This paper presents a self-supervised learning framework named BEIT, in which the input image can be masked in some regions, and the task is to recover the token of the masked region. Pre-trained on ImageNet, the model shows good performance on a series of downstream tasks.

**Summary Of The Review:**

1. The idea is good, going through an important direction of self-supervised learning.
2. Downstream tests show good performance.
3. BUT, whether the improvement comes from the designed framework is questionable.

I need additional details from the authors to make the final decision.

---

> ### Author Response · Authors · 2021-11-23
> **Response to Reviewer GYF1**
>
> Thanks for your feedback!
>
> First, I would like to clarify that the pre-training and the tokenizer do **not** rely on any image-text pairs, i.e., only unannotated images are used. So, the proposed framework is fairly compared. More details are described as follows.
>
> Q: the tokenizers are borrowed from DALL-E, which means that a large number of image-text pairs have been used. ... This is not considered self-supervised learning, as the texts can contain vast amount of semantics (according to the CLIP paper, after training on image-text pairs, the model can achieve good performance on zero-shot ImageNet classification)
>
> A: The [DALL-E tokenizer](https://arxiv.org/abs/2102.12092) does not use any image-text pair, which is learned by unsupervised learning with reconstruction loss. So, the whole framework is still self-supervised learning.
>
>
> Q: The detailed setting of the DALL-E tokenizer and ImageNet re-trained tokenizer. In particular, please specify what kind of external information has been used, and the cost of pre-training such tokenizers.
>
> A: The reimplemented tokenizer in Appendix C is re-trained on only ImageNet-1k images (without labels). We trained the tokenizer with 50 epochs (batch size=512; lr=5e-4) based on the [open-sourced repo](https://github.com/lucidrains/DALLE-pytorch) using 4 v100 GPU cards. The training cost of the above tokenizer is only about 1/33 of ViT-B pre-training, which is relatively cheap compared with the whole run. Moreover, the DALL-E tokenizer also does not use external information. The training of tokenizers uses reconstruction loss, which only needs images.
>
>
> Q: If external information is used
>
> A: No, we do not use any external information for pre-training. Please refer to the above answers.
>
>
> Q: a discussion on the relationship between this approach and knowledge distillation is necessary
>
> A: As shown in Table 4, the ablation study "-Masking +Recover 100% visual tokens'' (i.e., without masking and distill from the tokenizer) is equal to knowledge distillation with the tokenizer, which performs worse than fine-tuning from scratch (i.e., without pre-training). So, the key ingredient is masking and then recovering the original image from its corrupted version. Sec 2.4 helps to understand the proposed framework from the perspective of variational autoencoder.

---

> > ### Comment · Reviewer_GYF1 · 2021-11-24
> > **Response to the authors' rebuttal**
> >
> > Overall, I am satisfied with the rebuttal. I suggest the authors add the key information "We trained the tokenizer with 50 epochs (batch size=512; lr=5e-4), ... which is only about 1/33 of ViT-B pre-training ..." into the paper.

---

### Official Review · Reviewer_pM4q · 2021-11-01

**Correctness:** 4
**Technical Novelty And Significance:** 3
**Empirical Novelty And Significance:** 3
**Recommendation:** 8
**Confidence:** 4

**Main Review:**

I think the MIM objective resembles the masked region modeling (MRM) objective, which is widely used in the vision-and-language pretraining (VLP) models.
VLP models often mask several visual regions and make their contextualized outputs predict their corresponding object class to guide visual inputs in tandem with the textual inputs' masked language modeling.
By removing textual inputs from VLP models and switching detection-based region features to patches, we have a BEiT-like structure.
Recently, ViLT[^1] proposed this type of VLP model but failed to make appropriate objectives for the patches, reporting that regressing masked pixels (masked patch prediction objective) deteriorate downstream vision-and-language tasks.
In my view, as MRM and MIM are similar enough to be noted, I recommend the authors add the relation with BEiT and VLP models in the related work section.

Using argmax-ed visual tokens is inevitable for DALL.E since they had to plug the discrete tokens into the decoder.
However, since BEiT only uses the discrete tokens as ingredients of the MIM objective, I think argmax-ing the visual tokens is unnecessary, and the token distribution can be immediately used for the objective (e.g., KL-divergence).
Actually, UNITER[^2], which used detection-based regional inferred class for the MRM objective to train VLP model, has tested three types of using the class information:
1.  Use it as a one-hot label (as BEiT did).
2.  Use KL-div.
3.  Regress the features that are used to infer the class labels.

UNITER showed that the combination of (2) + (3) yields better performance than solely using (1).
I believe all three approaches are also available for BEiT and the discrete VAE; thus, I wonder whether they can further boost the performance of BEiT.

I believe using visual tokenizers such as discrete VAE can be a silver lining for the community and those seeking self-supervisable images' objectives, including the VLP community.
I think this paper showed rigorous and solid empirical results and well contributes to the community by providing valuable tools.

[^1] Kim, Wonjae et al. "ViLT: Vision-and-Language Transformer Without Convolution or Region Supervision." _ICML_ (2021).
[^2] Chen, Yen-Chun et al. "UNITER: UNiversal Image-TExt Representation Learning." _ECCV_ (2020).

**Summary Of The Paper:**

The paper presents a new objective called masked image modeling (MIM) to pre-train vision transformers, making the model predict a visual token from the masked patches.
The visual tokens are obtained by discrete VAE, which is trained on 250M images. (Though, the authors showed that training discrete VAE with only 1M images (imagenet-1K) is enough to demonstrate the power of the proposed objective.)
The authors pre-train the ViT with MIM and fine-tune the pre-trained weights (BEiT) on two visual downstream tasks: image classification and semantic segmentation, and showed superior performances compared to previous methods, including DINO and MoCo v3.
The authors also propose an additional trick called blockwise masking to improve BEiT further.

**Summary Of The Review:**

- Consider adding a VLP subsection to the related work section.
- There are several ways to exploit the discrete VAE tokens.
- The reviewer thinks the paper's results are solid.

---

> ### Author Response · Authors · 2021-11-23
> **Response to Reviewer pM4q**
>
> Thank you for the constructive comments.
>
> Q: recommend the authors add the relation with BEiT and VLP models in the related work section
>
> A: We will mention the explorations of VLP in the related work section as suggested.
>
>
> Q: several ways (1. use it as a one-hot label; 2. use KL-div; 3. regress the features that are used to infer the class labels) to exploit the discrete VAE tokens. all three approaches are also available for BEiT and the discrete VAE; thus, I wonder whether they can further boost the performance of BEiT.
>
> A: We tried KL-div (i.e., the second way) in our experiments, and the results became slightly worse compared with one-hot labels. My guess is that in UNITER such ways can learn object detection (OD) knowledge from the prediction distribution of the OD model, which is particularly important for vision-language tasks. We would try more settings as you suggested. Thanks for the suggestion!

---

### Official Review · Reviewer_Vvct · 2021-11-03

**Correctness:** 4
**Technical Novelty And Significance:** 3
**Empirical Novelty And Significance:** 3
**Recommendation:** 8
**Confidence:** 4

**Main Review:**

PROS:
- The task is interesting per se, as it brings the concept of BERT into Vision Transformers
- Well written paper and simple idea
- Good analysis

CONS:
- Some important details for practitioners are missing (e.g. choices on the masking)
- experiments of different models are sometimes difficult to compare

In general, I liked the paper a lot. However, I have some issues that should be answered and addressed before publication. I'll number them for convenience

1. Authors says that pre-training is run for 800 epochs, which is fine for large ML groups but it might be very demanding for smaller groups. Moreover, it does not help the comparison with all other SoTA methods. For example, DiNO is trained for 300 epochs. Moco v3 is an experimental paper, but the main results are obtained for 300 epochs (although they report 600 epochs as well). Now, I wonder two things: 1) are pre-training results in Table 1 obtained with 800 epochs for all the models? I mean, are they comparable? 2) I'd have liked to see a comparison between models at 300 epochs (as standard) in Table 1 or in another additional table. Appendix A might be in that direction but I did not understand it well as it is not well explained; 3) practitioners would really like to see the training curve of BEiT, to understand what budget should they invest to have the desired accuracy.

2. Since BEiT is somewhat based on BERT, why do you replace 40% of tokens? Is there a motivation behind this number? then, BERT task was to replace the token 80% of the time with a [MASK] token, 10% of the time with a random token, and 10% of the time keeping it as it was. Did you consider this strategy? As far as I know, this is beneficial for fine-tuning tasks.

3. It would be good to see the standard (small) datasets for downstream tasks such as CIFAR-10, Oxford Flowers-102 and Oxford Pets or cars alongside CIFAR-100. This would have made the paper more comparable. Moreover, why 150 epochs? I believe that the standard is 100.

4. Why is DINO in Table 10 with 400 epochs? why not 300 as standard?

5. It's good to have the appendix, but authors should refer to them and comment on the (interesting!) results in the main paper. E.g. Appendix E, F.

6. Are authors going to release the code?

MINOR:
- I would change the special token [S] to [CLS] as in BERT
- I would cite this recent paper in the intro talking about how Vision Transformers are data-hungry https://arxiv.org/abs/2106.03746 and some tricks for small datasets
- In Section 2.3 "pixel-level auto-encoding" might be confusing. I suggest rephrasing it and explaining it better (as it is in the intro).
- If you have time, for the camera ready, I'd love to see BeIT applied to some convolution transformers such as Swin or CvT
- I would like to see a comment on the throughput of DINO vs your paper in the main manuscript

**Summary Of The Paper:**

This paper presents a novel task called Masked Image Modeling (MIM), inspired by the more famous Masked Language Model task proposed by BERT in NLP. For this reason, the paper is called BEiT. BEiT relies on a pre-pre-trained tokenizer that transforms image patches into discrete tokens, which are then masked and predicted. Extensive experiments show that this self-supervised pre-training improve SoTA in various downstream tasks such as image classification and semantic segmentation.

**Summary Of The Review:**

The paper is well written, the idea is somewhat novel (novel in computer vision, less novel in general because of BERT). Experiments are good but improvable.

---

> ### Author Response · Authors · 2021-11-23
> **Response to Reviewer Vvct**
>
> Thanks for your feedback!
>
> Q: Are authors going to release the code?
>
> A: All the code, pretrained checkpoints, and scripts are available for reimplementation.
>
>
> Q: pre-training is run for 800 epochs, which is fine for large ML groups but it might be very demanding for smaller groups.
>
> A: Our ablation studies used 300 epochs for faster iterations, which is hardware-friendly for resource-demanding settings. Moreover, we do not store two copies of Transformer parameters for self-distillation/contrast, and do not use multi-crop augmentation. So our method is much more memory-efficient and faster than previous work.
>
>
> Q: are pre-training results obtained with 800 epochs for all the models? I mean, are they comparable? practitioners would really like to see the training curve of BEiT, to understand what budget should they invest to have the desired accuracy.
>
> A: Previous work used different settings. For example, dino-vit/base/16 used 400 epochs (as shown in their [configuration file](https://dl.fbaipublicfiles.com/dino/dino_vitbase16_pretrain/args.txt) and [training log](https://dl.fbaipublicfiles.com/dino/dino_vitbase16_pretrain/dino_vitbase16_pretrain_log.txt)),  iGPT was trained by 1M steps with 128 batch size (~105 epochs; because of their extremely large model size), and MoCo v3 reported 300/600 in their paper. If we consider the training throughput and cost, our method tends to be cheaper because we do not use multi-crop augmentation or two copies of Transformer parameters for self-distillation and contrastive learning. So the results are comparable in the sense of training cost. Moreover, as reported in MoCo v3's paper, "the gain of training longer is diminishing on ViT-B" for MoCo v3 (i.e., 600 epochs vs. 300 epochs). In contrast, our method can still benefit from training more epochs, which is also another advantage of our method, especially for self-supervised learning. We will add a curve to show the performance along with the number of training epochs in the camera-ready version as suggested.
>
>
> Q: Since BEiT is somewhat based on BERT, why do you replace 40% of tokens? Is there a motivation behind this number?
>
> A: The "information density" of a patch seems lower than one word in a text. So the masking ratio would be larger than 15% that is used for language data.
>
>
> Q: BERT task was to replace the token 80% of the time with a [MASK] token, 10% of the time with a random token, and 10% of the time keeping it as it was. Did you consider this strategy? As far as I know, this is beneficial for fine-tuning tasks.
>
> A: Thanks for your valuable suggestion! It makes sense to me. We would like to try this improvement.
>
>
> Q: (CIFAR-100) Why 150 epochs? I believe that the standard is 100.
>
> A: We used 100 for ImageNet-1k fine-tuning. Because the data size of CIFAR is much smaller, we enlarge its epoch number to 150. For a reference, DINO used 1000 epochs for CIFAR10 fine-tuning as shown in their [repo](https://github.com/facebookresearch/dino/issues/81#issuecomment-884935778).
>
>
> Q: Why is DINO in Table 10 with 400 epochs? why not 300 as standard?
>
> A: We report the DINO results by directly using their officially released checkpoints. As shown in the files of [training arguments](https://dl.fbaipublicfiles.com/dino/dino_vitbase16_pretrain/args.txt) and [logs](https://dl.fbaipublicfiles.com/dino/dino_vitbase16_pretrain/dino_vitbase16_pretrain_log.txt), the dino-vit/base/16 model was trained with 400 epochs. As there are too many hyperparams of DINO, we directly follow their best-reported checkpoint.
>
>
> Q: cite this recent paper in the intro talking about how Vision Transformers are data-hungry https://arxiv.org/abs/2106.03746 and some tricks for small datasets
>
> A: We added the citation as suggested. Thanks for the pointer.
>
>
> Q: In Section 2.3 "pixel-level auto-encoding" might be confusing. I suggest rephrasing it and explaining it better (as it is in the intro).
>
> A: We added the explanation of the term in Sec 2.3.
>
>
> Q: I would like to see a comment on the throughput of DINO vs your paper in the main manuscript
>
> A: We added it in the related work section.

---

> > ### Comment · Reviewer_Vvct · 2021-11-28
> > **thanks**
> >
> > I thank the authors for answering my review and for following my suggestions. However, I still think authors should release the model at lower epochs to help low-budgets groups to compare BEiT with other papers.
> >
> > > A: Previous work used different settings. For example, dino-vit/base/16 used 400 epochs (as shown in their configuration file and training log), iGPT was trained by 1M steps with 128 batch size (~105 epochs; because of their extremely large model size), and MoCo v3 reported 300/600 in their paper. If we consider the training throughput and cost, our method tends to be cheaper because we do not use multi-crop augmentation or two copies of Transformer parameters for self-distillation and contrastive learning. So the results are comparable in the sense of training cost. Moreover, as reported in MoCo v3's paper, "the gain of training longer is diminishing on ViT-B" for MoCo v3 (i.e., 600 epochs vs. 300 epochs). In contrast, our method can still benefit from training more epochs, which is also another advantage of our method, especially for self-supervised learning. We will add a curve to show the performance along with the number of training epochs in the camera-ready version as suggested.
> >
> > Thank you, but I also strongly suggest to the authors to release the model at 300 epochs, to allow other practitioners to compare the model at lower epochs
> >
> > > A: We used 100 for ImageNet-1k fine-tuning. Because the data size of CIFAR is much smaller, we enlarge its epoch number to 150. For a reference, DINO used 1000 epochs for CIFAR10 fine-tuning as shown in their repo.
> >
> > I strongly suggest the authors to show and release the fine-tuning at 100 epochs.

---

> > > ### Author Response · Authors · 2021-11-29
> > > **Response to Reviewer Vvct**
> > >
> > > The suggested checkpoints have been added to our release plan. Thank you!

---

### Official Review · Reviewer_7cfB · 2021-11-09

**Correctness:** 3
**Technical Novelty And Significance:** 3
**Empirical Novelty And Significance:** 4
**Recommendation:** 8
**Confidence:** 4

**Main Review:**

(+) The paper is clear and easy to follow.
(+) There are proper ablations for most of the design decisions.
(-)  The dependence on DALL-E to is a pretty hefty one (unsupervised pretraining on 250M images). This can potentially be lifted, but the authors made no attempt in this direction.
(+/-) To be fair with the above point, the authors did at least the ablation of predicting masked pixels directly (instead of visual tokens from DALL-E) in Table 4, and the performance drops by "only" ~1.7% top1 on ImageNet in the downstream task. This is still not great as this brings the accuracy below supervised training only, which is what is done as fine-tuning on this network in this case. This ablation should have been done in linear probing also.
(-) Regarding linear probes, the results (in Table 9) are somewhat disappointing. I wonder if this comes from the relatively small batch size (and/or lack of hyperparameters search) compared to other papers, or if there is a more fundamental reason for BEiT to be worse than contrastive methods there.
(=) No pretraining on larger than ImageNet-1K.
(+) The paper provides enough details for reproducing the results.



**Summary Of The Paper:**

This paper is one of the first to present state-of-the-art results for masked image modeling (BERT-style) self-supervised learning on images (contrastive approaches held the SOTA before). The whole system is based on a ViT encoder (e.g. with 16x16 pixels patches as input) to produce visual tokens. The pretraining objective consists in matching the visual tokens (one per image patch) from a provided visual tokenizer, with the ViT encoder. The tokenizer is a discrete VAE (with 8192 token types) from prior work called DALL-E (Ramesh et al., 2021). The results on ImageNet-1K are SOTA for comparable model sizes and pretraining data and settings. Idem for segmentation on ADE20K.

**Summary Of The Review:**

Overall, this is a strong paper presenting a significant improvement in unsupervised pretraining for images, that should be presented at ICLR. More analysis (e.g. linear probing) could make the paper stronger. The limits of the paper are fair to have, and can potentially be addressed in follow-ups.

---

> ### Author Response · Authors · 2021-11-23
> **Response to Reviewer 7cfB**
>
> Thanks for your feedback!
>
> Q: The dependence on DALL-E to is a pretty hefty one (unsupervised pretraining on 250M images). This can potentially be lifted, but the authors made no attempt in this direction.
>
> A: As shown in Appendix C, we re-train the image tokenizer on ImageNet-1K and compare it with the DALL-E tokenizer. Table 8 shows that our reimplemented tokenizer obtains comparable reconstruction loss and ImageNet fine-tuning performance compared with the off-the-shelf DALL-E tokenizer.
>
>
> Q: (linear probes) I wonder if this comes from the relatively small batch size (and/or lack of hyperparameters search) compared to other papers, or if there is a more fundamental reason for BEiT to be worse than contrastive methods there.
>
> A: The contrastive learning methods typically learn to aggregate the image-level features into a global vector, which is relatively suitable for linear probing. In contrast, the generative self-supervised learning methods, such as iGPT and ours, usually do not pretrain such global feature aggregation, which tends to make linear probes difficult.

---

### Decision · Program_Chairs · 2022-01-20

**Decision:**

Accept (Oral)

**Comment:**

Inspired by BERT and the corresponding masked language modeling objective, this paper proposes masked image modeling as a pre-training technique for vision transformer. More precisely, the image is tokenized using a pre-trained tokenizer, and the goal is to predict the token indices corresponding to masked patches of the image. As noted by the reviewers, the proposed method is simple, works very well in practice and the paper is well written. Since this work potentially opens a whole new research direction, my recommendation is to accept with oral presentation.